# Zinc Evaporation from Brass Scraps in the Atmosphere of Inert Gas

**DOI:** 10.3390/ma16145178

**Published:** 2023-07-23

**Authors:** Magdalena Wilk, Tomasz Matula, Leszek Blacha, Albert Smalcerz, Jerzy Labaj

**Affiliations:** 1Department of Metallurgy and Recycling, Faculty of Materials Science, Silesian University of Technology, Krasinskiego 8, 40-019 Katowice, Poland; wilkmagdalena@outlook.com (M.W.); leszek.blacha@polsl.pl (L.B.); 2Department of Industrial Informatics, Faculty of Materials Science, Silesian University of Technology, Krasinskiego 8, 40-019 Katowice, Poland; albert.smalcerz@polsl.pl; 3Department of Production Engineering, Faculty of Materials Science, Silesian University of Technology, Krasinskiego 8, 40-019 Katowice, Poland; jerzy.labaj@polsl.pl

**Keywords:** metal scraps, brass, metal evaporation process, thermogravimetry

## Abstract

A description of the process of metal evaporation from liquid alloys at an atmospheric pressure has a practical value for both the smelting and remelting of their scraps. The quantities of volatile components that are eliminated in these processes depend on many factors of which the type of melting device, the method and conditions of the process performance, the alloy composition and the kind of applied atmosphere are of the greatest importance. In this paper, the results of the research on zinc evaporation from brass scraps containing 10.53 wt% Zn are presented. The experiments were conducted using the thermogravimetric method at 1080 ÷ 1240 °C in a helium atmosphere. In the research, the levels of zinc removal from copper ranged between 82% and 99%. The values of the overall mass transfer coefficient for zinc *k_Zn_*, determined based on the experimental data, ranged from 4.74 to 8.46 × 10^−5^ ms^−1^. The kinetic analysis showed that the rate of the analysed process was determined by mass transfer in the gas phase.

## 1. Introduction

Copper alloys belong to the group of metallic non-ferrous materials that are widely used in various economic sectors, which mainly results from their high strength and corrosion resistance. A growing demand for these materials requires more extensive activities aimed at recovering the basic components of these alloys from various types of metallic wastes or scraps. A higher interest in secondary copper is mostly associated with the lower energy costs related to its production compared to primary copper extracted from ores. When copper is produced from scraps, up to 80% of energy necessary for the production of primary copper is saved [1,2]. In addition, while comparing technologies of copper production from primary and secondary raw materials, more negative effects on the environment are observed for the primary materials, which is mainly associated with the processes of ore extraction and the pyrometallurgical processes of copper production. In the case of secondary copper, the major operations that affect the natural environment are refining processes. The available data from the literature show that the overall negative effects of secondary copper production on the environment do not exceed 15% of the estimated value for primary copper production [3]. Due to the fact that secondary copper is obtained both from post-production and from various types of post-amortisation scraps, its refining operations become particularly important to ensure proper chemical purity of the produced metallic materials. There are many papers that present the findings of research on zinc recovery from various kinds of cupriferous wastes and copper scraps. Cupriferous wastes are, e.g., casting waste, copper smelting dusts or flotation tailings. In this area, the offered solutions with regard to zinc removal from these wastes are primarily based on pyrometallurgical [4,5,6,7,8,9,10,11,12,13,14,15,16] and hydrometallurgical [17,18,19,20,21,22,23,24,25,26,27,28,29] processes. In hydrometallurgical processes, various leaching materials are applied such as sulphuric acid, hydrochloric acid, acetic acid or ammonia solutions. It should be noted, however, that hydrometallurgical methods are based on complex processing systems, which very frequently generate high operating costs. A number of research results are available in the literature, characterizing the research area related to the recognition of the metal evaporation phenomena, especially regarding zinc. This may indicate the importance of the topic with which research teams around the world are dealing. The authors of [30] presented the results of studies on the evaporation of zinc from zinc-bearing waste generated in the process of galvanizing steel surfaces. The process was carried out at a pressure of 100 kPa in a helium atmosphere and a process temperature of 700 to 800 °C. The result of these tests was obtaining zinc with a purity of 99 wt.%. In turn, in the case of research on the use of a vacuum in obtaining super pure zinc, the authors of work [31] obtained, as a result of multi-stage processing at a pressure of 0.28 Pa, m6N purity zinc. In the case of wastes such as yellow phosphorous flue dust (YPFD) containing zinc, the authors of paper [32] conducted studies on the recovery of zinc using vacuum metallurgy. The tests were carried out at a temperature of 900 °C and a pressure of 5–10 Pa. In this case, the recovery of Zn was 92.47 wt.%, and the recovery of lead was at the level of 99.78% by weight. The authors indicated that the YPFD vacuum processing technology is highly effective and environmentally friendly. The problem of processing EAFD zinc-containing electrosteel process waste is presented in [33]. The authors focused on determining the activation energy of the reduction process at different temperatures and at a pressure of 20 Pa. They determined the value of energy for the substrate transport process, the actual process and the diffusion process in the gas phase. In the study of the process of carbochemical dust reduction from the shaft furnace process, the authors of paper [34] carried out a theoretical analysis of the vacuum distillation process. The tests carried out in the temperature range of 800–900 °C enabled the reduction of zinc in the processed material by 99.6% by weight. In addition, phases containing iron and zinc present in the tested material were identified, and the course of the reduction process was indicated. In [35], the authors present the results of the research on the refining process of blister copper containing a number of impurities that are characterized by a higher vapor pressure than the base metal. The tests were carried out in the temperature range of 1200–1250 °C at pressures from 8 to 533 Pa. Based on the determination of the kinetics of the component evaporation process, the stages determining the process of removing elements depending on the pressure level in the test system were defined in the work. In paper [36], the authors, using the example of a copper alloy, presented the possibility of using VIM technology to carry out the process of removing components via evaporation. The obtained results indicate that the use of the phenomenon of inductive mixing of the liquid bath significantly affects the efficiency of the process by increasing the mass transport coefficient in the liquid phase. In paper [37], the phenomenon of the evaporation of liquid alloy components at a reduced pressure is presented in a theoretical and practical way. In this case, the authors explain the phenomenon of the evaporation of manganese from liquid iron. In this process, the parameter of the total mass transport coefficient was determined, which increases from 3.9·10^−6^ ms^−1^ to 208.4·10^−6^ ms^−1^ with the pressure reduction in the system from 1000 Pa to 10 Pa. Paper [38] presents the result of the research on the process of the vacuum refining of iron in order to reduce the copper content. Based on the obtained results, the authors determined the parameters characterizing the mass transport processes in the individual phases of the system. The test results confirm the possibility of lowering the copper content in iron below the level of 0.3 wt.%. The authors of paper [39] present the results of high-temperature remelting tests in terms of characterizing mass transport via diffusion. The tests were carried out at a temperature of 1600 °C. The test results are highly correlated with those determined theoretically. In [40], the authors present the results of the evaporation process of a number of elements in He and N_2_ atmospheres, which indicate a strong dependence on the pressure in the system. The dependence of the content of oxygen and water (in the inert gas) on the increase in evaporation of the tested elements in the oxide form was also indicated. Papers [41,42] present the possibility of separating components during processing in a vacuum system equipped with a sublimation system. The results of work [42] indicate a high degree of zinc removal, even up to 96.09 wt.%, at a temperature of 1050 °C and a pressure of 50 Pa. On the other hand, at a pressure of 500 Pa, the value of zinc removal was 94.73% by weight.

In this paper, the results of the research on zinc removal from liquid copper via its evaporation in the atmosphere of inert gas are presented. Analysing the zinc evaporation from liquid copper alloy is a complex process due to its heterogeneous nature. Contrary to processes performed in homogeneous environments, the components of various phases in the system may only react with each other after they are transferred to the interface. In this case, the resultant rate of the process is affected by the mass transfer phenomena in addition to the chemical affinity. The evaporation rate of a component of a liquid multi-component alloy is mainly controlled by mass transfer processes, both in the gas phase and in the liquid phase. Changes in the temperature, pressure and hydrodynamic conditions in the system may significantly affect the rate of the process [43,44,45,46,47]. In addition, the chemical composition of the alloy (particularly its surface active components) may change the evaporation rate. To discover which of these factors most markedly affects evaporation kinetics as well as to determine the nature of this process control require experimental studies.

## 2. Materials and Methods

In the experiments, Cu–Zn alloy scrap was used (see Table 1 for its chemical composition). The research material consisted of rings that are waste elements of pipe production (Figure 1). In each of the measurements, the weight of the sample was about 1 g, and the particle size was in the range of 4 ÷ 6 mm.

For the experiments, the STA 449 F3 Jupiter thermal analyser, Netzsch (Selb, Germany), was applied (see Figure 2).

The device contains a graphite furnace operating in a protective environment in the presence of a selected gas or gas mixture. During the operation, the furnace shell is constantly cooled by water that is delivered through an open system. Before the experiment, an alloy sample of a particular weight was placed inside a small DTA/TG Al_2_O_3_ crucible, which was then attached to the measuring head in the working chamber of the analyser. The following parameters were recorded during the experiments: sample mass loss, temperature, experiment duration and gas flow rate. All measurements were performed in a helium atmosphere at the flow rate of 50 mL/min. The heating program selected on the basis of the melting temperature of the tested alloy (Figure 3) consisted of the following three essential stages:Sample heating up to a desired temperature (1080, 1120, 1160, 1200, 1240 °C) at 20 °C/min;Isothermal holding of the sample for 30 min at the set temperature;Sample cooling to 900 °C.

### Equilibrium Pressure of Zinc over Cu–Zn Alloys

While analysing a given process (1), it should be remembered that the evaporation of a volatile alloy component is only possible when its vapour pressure in specified conditions is markedly higher than the vapour pressures of the other alloy components. A decreased content of the specific alloy component results in a smaller difference between its vapour pressure and the vapour pressures of the other components. This means that elimination of this component of the liquid metal via its evaporation is only possible up to a specific limited value, and when this value is nearly reached, the other alloy components also begin to evaporate intensely. When the chemical potentials of the component “*i*” of the liquid phase (liquid metal alloy) and of the gas phase are equal, this may serve to calculate the vapour pressure of this component as follows:(1)$$Zn_{\left(l\right)\;}=Zn_{\left(g\right)}$$
(2)$$\mu_{i\left(l\right)\;}=\mu_{i\left(g\right)}$$
(3)$$\mu_{i\left(l\right)\;}=\mu_{i\left(g\right)}^{0}+RT\ln p_{i}\;$$
where:$Zn_{\left(l\right)\;}$*,*$Zn_{\left(g\right)}$—the zinc in the liquid phase and in the gas phase, respectively;$\mu_{i\left(l\right)}^{0}$, $\mu_{i\left(g\right)}^{0}$—the standard chemical potentials of the pure component “*i*” in the liquid and gas phases, respectively;$p_{i}$—the partial equilibrium pressure of the component “*i*” over the solution.

The latter parameter is determined as follows:(4)$$p_{i}=\gamma_{i}\cdot X_{i}\cdot p_{i}^{0}$$
where:$X_{i}$—the mole fraction of the component “*i*” in the alloy;$p_{i}^{0}$—the vapour pressure of the component “*i*” over the pure liquid.

Equation (4) proves that in order to determine the vapour pressure of a liquid metal solution component, its vapour pressure over the pure liquid and the activity coefficient of this component in the solution must be known.

Based on the above equations, the values of zinc and copper activity coefficients over liquid Cu–Zn alloys were determined. The following equations (recommended by Plewa in [47] and formulated based on Gerling and Predel’s data [48]) were used:(5)$$ln\gamma_{Cu}=\left(-\frac{5120}{T}+1.61\right)\left[X_{Cu}^{2.48}\right]$$
(6)$$ln\gamma_{Zn}=\left(-\frac{5120}{T}+1.61\right)\left[(X_{Cu}^{2.48}-1.68)\left(X_{Cu}^{1.48}+0.68\right)\right]$$

## 3. Results and Discussion

The thermodynamic data from HSC Chemistry 10.3.4 were applied to determine the vapour pressures of copper $p_{Cu}^{0}$ and zinc $p_{Zn}^{0}$ over the pure liquid [4]. The changes in the vapour pressures of zinc and copper over the alloys containing up to 11 wt% Zn within 1080 ÷ 1240 °C are presented in Figure 4 and Figure 5, respectively. The data show that the estimated values *p_Cu_* and *p_Zn_* are markedly different; at 1080 °C, *p_Cu_* is 0.02 Pa and *p_Zn_* is 39,529 Pa, while at 1240 °C, they are 0.14 Pa and 83,000 Pa, respectively. The *p_Zn_/p_Cu_* ratio values for the studied alloy (10.53 wt% Zn) within the analysed temperature range are above 5 × 10^5^ °C(Figure 6). Thus, it may be assumed that practically only zinc is present in the gas phase over the liquid Cu–Zn alloy in the temperature conditions of the experiments.

The study conducted based on the above research plan yielded a series of thermogravimetric (TG) curves. To analyse the results, examples of TG curves for the experiments performed at 1080, 1160 and 1240 °C are presented in Figure 7, Figure 8 and Figure 9.

The above TG curves were used to determine the mass losses of the specific samples during their heating and isothermal holding for each experiment (Table 2). Based on Equation (7) and the changes in the sample masses observed in the TG curves, the level of zinc removal from the alloy was estimated. The results of the calculations are also presented in Table 2.
(7)$$C_{Zn}=\frac{m_{0}-m_{k}}{m_{0}}\cdot 100\%$$
where:m_0_—the initial sample mass;*m_k_*—the final sample mass.

In addition, the changes in the zinc concentration in the alloy during the experiments were estimated based on the results of the thermogravimetric measurements. The mass losses of the remaining elements will not have a significant impact on the calculations of the metal removal rate in this case. The exemplary changes in the zinc content in the alloy during the experiments performed at 1080 °C and 1160 °C are presented in Figure 10.

One of the issues that are necessary to determine the reaction kinetics is the determination of its driving force ∆π, which is the difference between the vapour concentrations of metal in the liquid phase and in the gas phase. The driving force ∆π for the analysed system is generally expressed as the difference between the concentrations of zinc in the liquid phase (liquid Cu–Zn alloy) and in the gas phase (helium). The analysed process is an example of a process of zinc mass transfer through inerts. For this process, the value of ∆π is determined using the following equation (Hobler):(8)$$\Delta \pi =\ln\left(\frac{1}{1+Y_{Zn}^{*}}\right)$$
where:$Y_{Zn}^{*}$—the equilibrium concentration of zinc over the liquid Cu–Zn alloy.

The value $Y_{Zn}^{*}$ can be estimated with the use of the following equation:(9)$$Y_{Zn}^{*}=\frac{p_{Zn}}{P-p_{Zn}}$$
where:*P*—the overall pressure in the system.

While analysing the mass losses of the specific samples within the isothermal range of the measurements, it was found that the increasing temperature caused their decrease, which is a result of the change in the process driving force.

Figure 11 illustrates the values ∆π = f(C_Zn_) for the analysed evaporation process, determined based on Equation (8). It can be clearly seen that when the content of zinc in the alloy decreases, the value of the driving force also becomes reduced, which explains the fact that the temperature rise resulted in smaller mass losses of the samples within the isothermal range of the measurements.

As previously noted, analysing the zinc evaporation from the liquid copper alloy is a complex process due to its heterogeneous nature, and its resultant rate is affected by the mass transfer phenomena both in the liquid phase and in the gas phase in addition to the chemical affinity. Kinetically, this process may be divided into the following three essential stages [30]:Zinc transfer from deep inside of the liquid alloy to the interface;Zinc evaporation from the liquid metal–gas phase interface;Zinc vapour transfer from the interface into the deeper part of the gas phase.

Mass transfer in the liquid and gas phases in the hydrodynamic conditions that are typical of metallurgical aggregates primarily occurs via convection and, to a lesser extent, via diffusion.

Based on the above assumptions, the value of the overall zinc mass transfer coefficient *k_Zn_* in the analysed process of evaporation from the copper alloy can be determined with the use of the following equation:(10)$$\frac{1}{k_{Zn}}=\frac{1}{k^{l}}+\frac{1}{k^{e}}+\frac{1}{k^{g\;}}$$
where:*k^l^*—the mass transfer coefficient of Zn in the liquid phase;*k^g^*—the mass transfer coefficient of Zn in the gas phase;*k^e^*—the Zn evaporation rate coefficient.

As proven in many studies, the value of the overall mass transfer coefficient *k_Zn_* can be estimated based on experimental data when the change in the concentration of the component evaporating from the liquid vs. the time is known [49]. Assuming that the evaporation of a component of a liquid metal alloy can be generally described by a first-order kinetic equation, the following equation was used to determine the coefficient *k_Zn_* based on the experimental data:(11)$$2.303\log\frac{C_{Zn}^{t}}{C_{Zn}^{0}}=-k_{Zn}\frac{F}{V}\left(t-t_{0}\right)$$
where:
$C_{Zn}^{0}\;and\;C_{Zn}^{t}—$the initial zinc content in copper and its content after time t, wt%;*F*—the evaporation surface, m^2^;V—the volume of the liquid copper alloy, m^3^;(t – *t_0_)*—the duration of the process, s.

For the melting device used in the experiments, the determination of the values of the zinc transfer coefficient in the liquid phase *k^l^* and in the gas phase *k^g^* was not possible. In the performed analysis, only the rate constant *k^e^* could be estimated, using the following equation [50]:(12)$$k^{e}=\frac{\alpha \cdot \;p_{Zn\;}^{0}\;\gamma_{Zn}\cdot \;M_{Cu}}{{\left(2\pi RTM_{Zn}\right)}^{0.5}\;\cdot \;\rho_{Cu}}$$
where:α—the evaporation constant; *M_Cu_, M_Zn_*—the molar masses of copper and zinc, respectively;*ρ_Cu_*—the copper density.

The determined values of the *k_Zn_* and *k_e_* are summarised in Table 3.

An important parameter that determines the rate and the course of the process is its resistance. In this paper, the calculations of resistance related to the reaction on the surface of the liquid metal and its fraction in the total resistance of the process are presented. To determine the fraction of resistance related to the reaction on the surface of the liquid alloy (*R^e^*) in the total resistance of the analysed evaporation process, the following equation was used:(13)$$R^{e}=\frac{\left(\frac{1}{k^{e}}\right)}{\left(\frac{1}{k_{Zn}}\right)}\cdot 100\;\%$$

Based on the performed analyses, it was demonstrated that the fraction of resistance *R^e^* in the total resistance of the process ranged from 10% to 18%. This means that evaporation itself, which occurs on the surface of the liquid alloy, is not a stage that limits the rate of the analysed process.

Figure 12 illustrates the mass transfer coefficient *k_Zn_* vs. the temperature in the coordinate system (Arrhenius).

This method of interpreting the results based on Equation (14) allowed for the determination of a hypothetical value of activation energy for the studied process of zinc evaporation.
(14)$$E_{A}=-RT\ln\left(\frac{k}{A}\right)$$
where:*k*—the reaction rate constant, m/s;*A*—the constant for the specific reaction, m/s;*R*—the gas constant, J/mol·K;*T*—the temperature, K.

The value is 68.90 kJ·mol^−1^. For comparison, the value of activation energy for zinc diffusion in liquid copper, estimated based on the values of coefficients of zinc diffusion in liquid copper, is 18 kJ·mol^−1^. The significant difference between these activation energy values shows that the process of zinc transfer in liquid copper in the applied measurement system does not determine the zinc evaporation rate.

However, it should be noted that even in processes that are clearly characterised by diffusion control, the values of activation energy determined based on the overall mass transfer coefficients do not have to be identical with the diffusion activation energy. This depends on the hydrodynamic conditions in the measurement system. Due to the fact that diffusion in the gas phase is not an activated process, it was not possible to compare the diffusion activation energy in this phase with the estimated values of the hypothetical activation energy for the studied process of zinc evaporation.

The performed analysis of the research results shows that the analysed process of zinc evaporation from the Cu–Zn alloy performed in the atmosphere of inert gas was characterised by diffusion control, and its rate was determined via mass transfer in the gas phase. Similar conclusions were drawn from the data in the literature presented in paper [49].

In paper [51], the results of the research on the rate of zinc evaporation from Cu–Zn alloys in the atmospheres of argon, carbon monoxide and helium are presented. Examples of changes in the zinc concentration in the alloy containing copper during the remelting process (obtained by the authors) are shown in Figure 13.

The data presented in this figure show that the rate of the process of zinc evaporation was far higher in the helium atmosphere compared to the carbon monoxide or argon atmospheres, and this rate was higher in the carbon monoxide atmosphere compared to the argon atmosphere. It should be noted that the coefficients of zinc vapour diffusion in the analysed gas atmospheres are arranged in a series similar to that of the estimated values of the evaporation rate as follows:*D_Zn-He_* > *D_Zn-CO_* > *D_Zn_-_Ar_*(15)

In paper [40], the effects of the type of applied gas atmosphere on the rates of molybdenum evaporation in helium, argon and nitrogen are demonstrated. Here, the highest evaporation rate was also achieved in the helium atmosphere, which is illustrated in Figure 14. In this figure, the evaporation rate is expressed as the ratio to the maximum rate observed in a perfect vacuum.

## 4. Conclusions

Based on the thermodynamic and kinetics analysis of the Cu–Zn system and the obtained results of the thermogravimetric study of the evaporation process, it was possible to formulate the following conclusions:The estimated zinc vapour pressure over the analysed Cu–Zn alloy at 1080 ÷ 1240 °C ranges from 8.80 × 10^8^ Pa to 1.19 × 10^9^ Pa. These values range from 5.12 × 10^3^ Pa to 5.28 × 10^4^ Pa for copper. The significant difference in the vapour pressures of both metals that form the alloy allows for the assumption to be made that during the liquid phase, practically only zinc is evaporated.For the assumed temperature range and the atmospheric pressure (the conditions of the experiments), the achieved level of zinc removal from the Cu–Zn alloy ranges from 83% to 99.9%.Based on the determined kinetic parameters of the zinc evaporation process, i.e., the overall zinc mass transfer coefficient *k_Zn_* and the evaporation rate constant *k_e_*, it can be concluded that the analysed process is characterised by diffusion control and the stage that determines its rate is the mass transfer in the gas phase.The estimated value of apparent activation energy for the studied process of zinc evaporation is 69 kJ·mol^−1^. For comparison, the value of zinc diffusion activation energy in liquid copper, estimated based on the coefficients of diffusion of these metals, is 18 kJ·mol^−1^.For all experiments, the determined fraction of resistance related to the process on the liquid alloy *R_e_* in the total resistance of the process ranges from 10% to 18%, which means that evaporation itself, which occurs on the interface, is not a factor that limits the rate of the analysed process.

## Figures and Tables

**Figure 1 materials-16-05178-f001:**
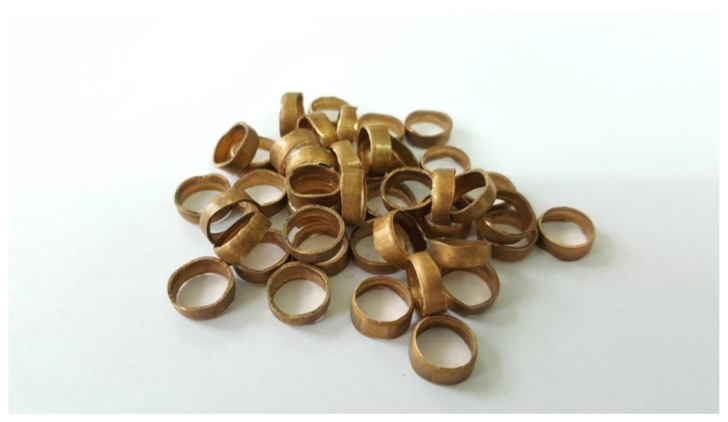
Research material.

**Figure 2 materials-16-05178-f002:**
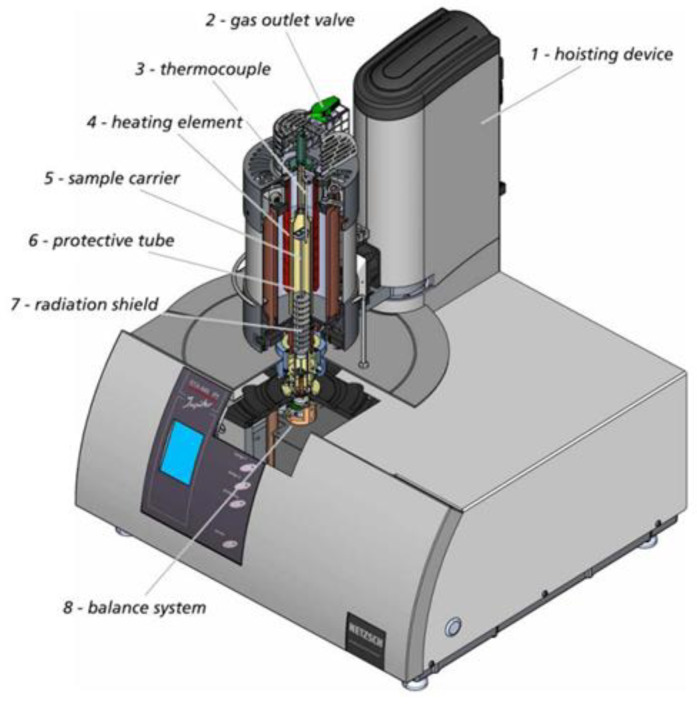
Components of the STA 449 F3 Jupiter thermal analyser.

**Figure 3 materials-16-05178-f003:**
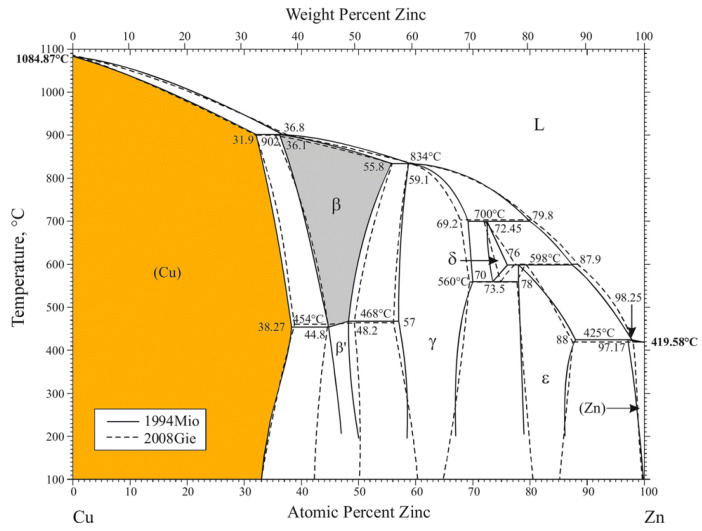
Cu-Zn binary phase diagram [46].

**Figure 4 materials-16-05178-f004:**
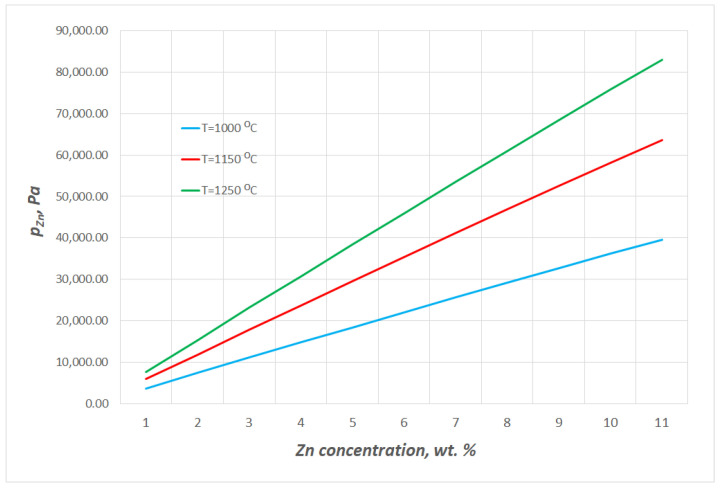
Selected changes in equilibrium vapour pressures of zinc over liquid Cu–Zn alloys containing up to 11 wt% Zn.

**Figure 5 materials-16-05178-f005:**
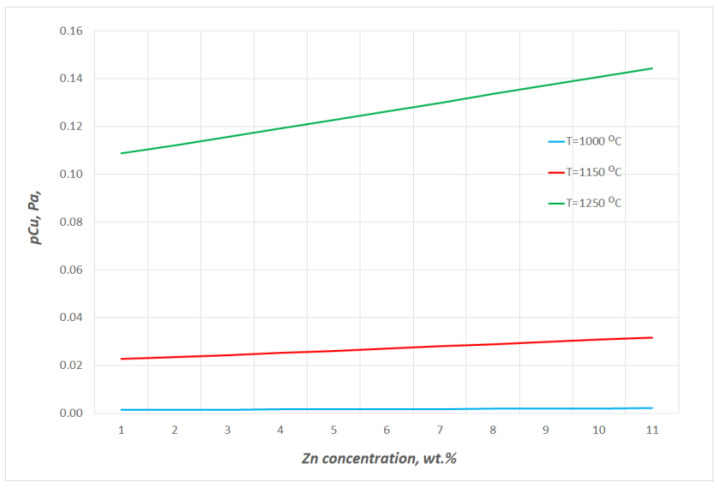
Selected changes in equilibrium vapour pressures of copper over liquid Cu–Zn alloys containing up to 11 wt% Zn.

**Figure 6 materials-16-05178-f006:**
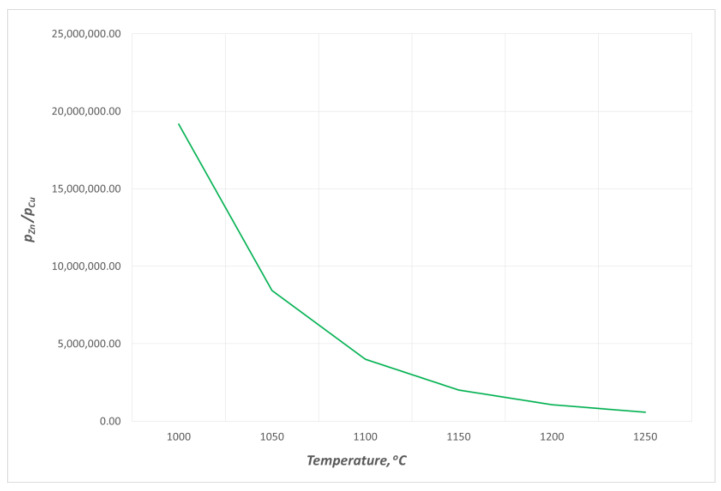
Selected changes in the ratio values of equilibrium vapour pressures of zinc and copper over liquid Cu–Zn alloys for the alloy containing 10.53 wt% Zn.

**Figure 7 materials-16-05178-f007:**
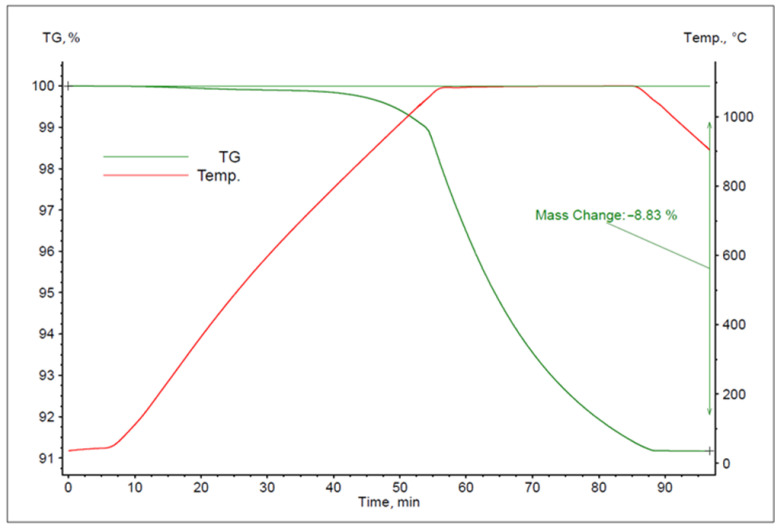
The TG curve for the experiment performed at 1080 °C.

**Figure 8 materials-16-05178-f008:**
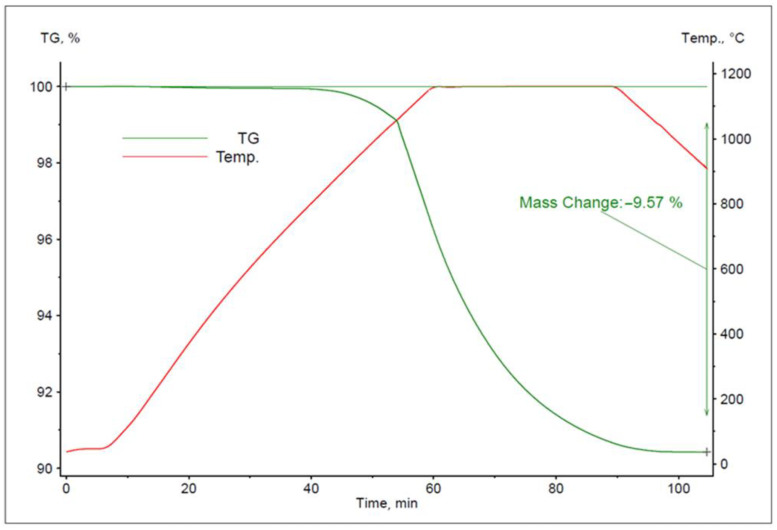
The TG curve for the experiment performed at 1160 °C.

**Figure 9 materials-16-05178-f009:**
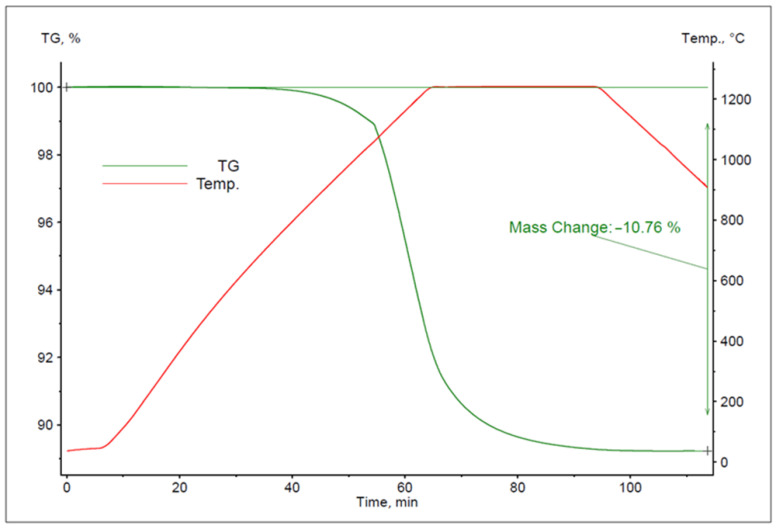
The TG curve for the experiment performed at 1240 °C.

**Figure 10 materials-16-05178-f010:**
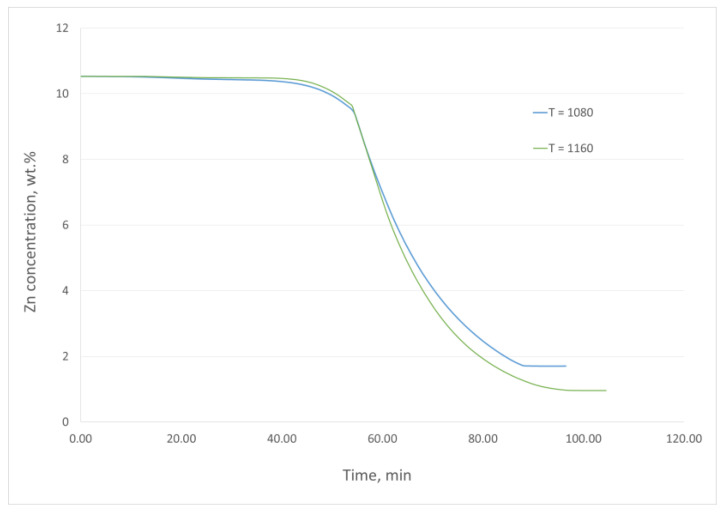
Estimated changes in zinc concentration in the alloy during the experiments performed at 1080 °C and 1160 °C.

**Figure 11 materials-16-05178-f011:**
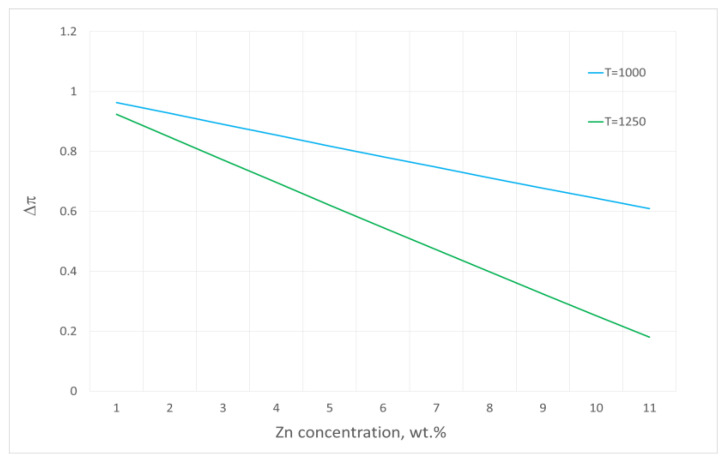
Changes in the driving force values in the process of zinc evaporation from the studied Cu–Zn alloy for 1000 °C and 1250 °C.

**Figure 12 materials-16-05178-f012:**
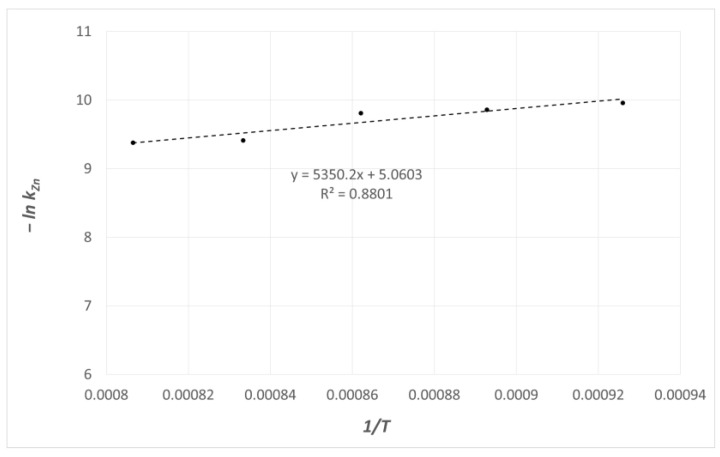
The mass transfer coefficient *k_Zn_* vs. temperature.

**Figure 13 materials-16-05178-f013:**
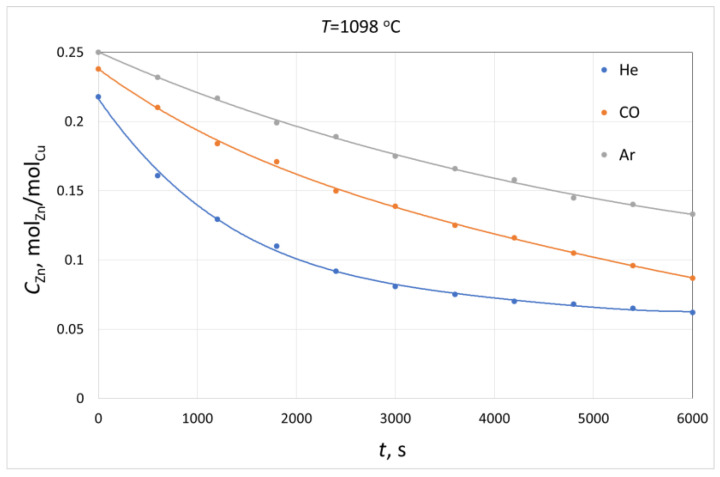
Changes in zinc concentration in the Cu–Zn alloy during its smelting in various gas atmospheres, T = 1098 °C, *C*^0^*_Zn_* = 0.252 moleZn/moleCu.

**Figure 14 materials-16-05178-f014:**
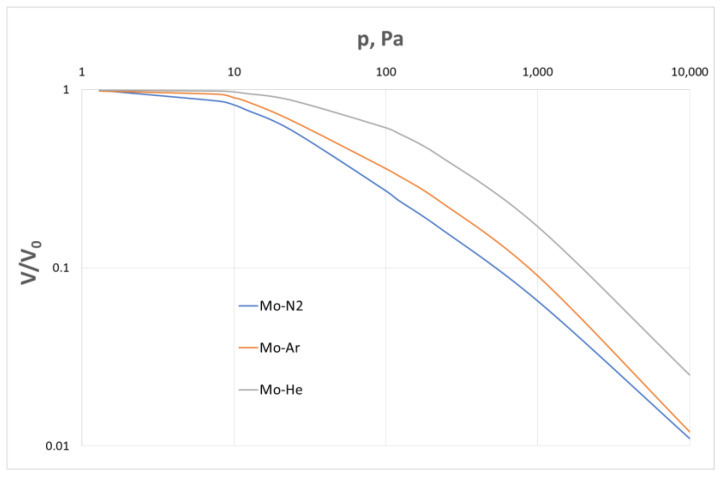
The effects of argon, helium and nitrogen pressures in the measurement system on the molybdenum evaporation rate.

**Table 1 materials-16-05178-t001:** Chemical composition of the Cu–Zn alloy.

Component	Zn	Fe	Al	Ni	Sn	Cu
Content, %	10.53	0.18	0.01	0.3	0.08	balance

**Table 2 materials-16-05178-t002:** Summarised levels of zinc removal from the alloy based on the values of sample mass changes during the process.

No.	Temperature, °C	Sample Mass Loss during Its Heating, mg	Sample Mass Loss during Its Isothermal Holding, mg	The Total Weight Loss of the Sample during the Heating Process, mg	Level of Zn Removal, %
1	1080	18.67	69.41	89.64	83.82
2	1120	28.03	65.67	95.71	89.88
3	1160	38.06	56.71	96.74	90.90
4	1200	57.40	45.89	103.89	98.19
5	1240	80.09	27.90	108.30	99.87

**Table 3 materials-16-05178-t003:** The overall mass transfer coefficient *k_Zn_* and coefficient *k^e^_Zn_*.

T, °C	*k_Z_*_n_, m s^−1^	*k^e^*, m s^−1^
1080	4.74 × 10^−5^	4.72 × 10^−4^
1120	5.22 × 10^−5^	4.65 × 10^−4^
1160	5.48 × 10^−5^	4.58 × 10^−4^
1200	8.18 × 10^−5^	4.52 × 10^−4^
1240	8.46 × 10^−5^	4.46 × 10^−4^

## Data Availability

Data sharing is not applicable to this article.

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
