# Peer review of "Zinc Evaporation from Brass Scraps in the Atmosphere of Inert Gas"

_materials, 2023, doi:10.3390/ma16145178_

Round 1
Reviewer 1 Report
Authors investigated the zinc evaporation from brass scraps in the atmosphere of inert gas. The following corrections/revisions essentially required to be addressed as given below:
(1) Add the source of research material.
(2) Add the Cu-Zn binary phase diagram or the curve of melting point of Cu-Zn alloy vs Zn concentration to indicate the reasonability of selected isothermal holding temperatures.
(3) Tables are not wrongly numbered.
(4) What is the difference between the third column and fifth column of Table in Page 7? Add the initial sample mass and final sample mass in the Table.
(5) There are no experimental data at 1000℃ and 1250℃. How the data of Δπ are calculated in Figure 10? In especial, the melting point of Cu-Zn alloy are larger than 1000℃ with low zinc concentration.
Authors investigated the zinc evaporation from brass scraps in the atmosphere of inert gas. The following corrections/revisions essentially required to be addressed as given below:
(1) Add the source of research material.
(2) Add the Cu-Zn binary phase diagram or the curve of melting point of Cu-Zn alloy vs Zn concentration to indicate the reasonability of selected isothermal holding temperatures.
(3) Tables are not wrongly numbered.
(4) What is the difference between the third column and fifth column of Table in Page 7? Add the initial sample mass and final sample mass in the Table.
(5) There are no experimental data at 1000℃ and 1250℃. How the data of Δπ are calculated in Figure 10? In especial, the melting point of Cu-Zn alloy are larger than 1000℃ with low zinc concentration.
Author Response
Thank you for your review.
Authors investigated the zinc evaporation from brass scraps in the atmosphere of inert gas. The following corrections/revisions essentially required to be addressed as given below:
(1) Add the source of research material.
Thank you for your comment. The source of research material is mentioned in the text in the second paragraph.
(2) Add the Cu-Zn binary phase diagram or the curve of melting point of Cu-Zn alloy vs Zn concentration to indicate the reasonability of selected isothermal holding temperatures.
Thank you for your comment. In accordance with the comments of Reviewer 1, the Cu-Zn phase system was added, which justifies the selection of the temperature range in which the measurements were carried out.
(3) Tables are not wrongly numbered.
Thank you for your comment. Table numbering has been corrected.
(4) What is the difference between the third column and fifth column of Table in Page 7? Add the initial sample mass and final sample mass in the Table.
Thank you for your comment. The column title note has been taken into account and the text has been corrected. The text shows the initial weight of the samples, while the fifth column in Table 2 shows the total weight loss of the samples. The authors decided that adding one more column with the final weight of the sample is unjustified.
(5) There are no experimental data at 1000℃ and 1250℃. How the data of Δπ are calculated in Figure 10? In especial, the melting point of Cu-Zn alloy are larger than 1000℃ with low zinc concentration.
Thank you for your comment. The value of pi was estimated (not calculated from the test results) based on the initial concentration and a slightly wider range of temperatures used in the tests to show the trend of changes in this parameter.

Reviewer 2 Report
Dear authors,
It is very difficult to review the manuscript without line numbers. Therefore, I have added my comments in the PDF file.

Author Response
Thank you for your review.
All shortcomings have been corrected. All changes are presented in the comments in the attached file.

Reviewer 3 Report
The study aimed the zinc separation by evaporation. In the recycling scenario, this study is important and may bring new industrial processes. In academic scenario, the novelty is not well described.
The lines are not numbered.
In Introduction section, the authors states that “There are many papers presenting the findings of research on zinc recovery from various kinds of cupriferous wastes and copper scraps”. But the authors did not present any reference or discussion regarding this. The state of art should be presented.
In Table 1, what is the Cu content? “Balance” means nothing, and a value should be presented.
Please, avoid short paragraphs as in page 2.
Figures 3-5 should be presented in the Results section.
Figure 11: the correlation is too low.
I recommend the authors to present the conclusions numbered instead of bullet points.
There is no chemical equation presented to explain the experimental data.
Author Response
Thank you for your review.
All shortcomings have been corrected. All changes are presented in the comments in the attached file.
- The lines are not numbered.
The article was prepared in Journal template. There were no numbered lines in it.
- In Introduction section, the authors states that “There are many papers presenting the findings of research on zinc recovery from various kinds of cupriferous wastes and copper scraps”. But the authors did not present any reference or discussion regarding this. The state of art should be presented.
Thank you for pointing this problem. In response to the comment, the number of cited papers was corrected in terms of topics, the subject of which broadly covered the area of phenomena related to the process of metal evaporation. The cited literature items, which focus on the preservation of zinc in systems in which the conditions and possibilities of zinc evaporation are determined, have been retained.
- In Table 1, what is the Cu content? “Balance” means nothing, and a value should be presented.
Thank you for bringing this term to our attention. The authors used the term "balance" in accordance with the applicable standards for determining the chemical composition of the alloy, in which the main component of the alloy is always referred to as "balance" or "residual". Such terms are used in all standards specifying the chemical composition of the alloy.
- Please, avoid short paragraphs as in page 2.
Thank you very much for drawing attention to the problem. The problem has been corrected in the article.
- Figures 3-5 should be presented in the Results section.
Thank you very much for drawing attention to the problem. The problem has been corrected in the article.
- Figure 11: the correlation is too low.
Following the comments of reviewer, R2 factor was added to show that the correlation is not low.
- I recommend the authors to present the conclusions numbered instead of bullet points.
Following the comments of reviewer, the presentation of the conclusions was changed from bullet points to numbered points
- There is no chemical equation presented to explain the experimental data.
Thank you very much for drawing attention to the problem. A chemical equation (1) explaining the experiment was added to describe the zinc evaporation process.

Round 2
Reviewer 2 Report
Dear authors,
Your manuscript has undergone significant improvements after revision and can be published with minor revisions. It would be beneficial to enhance the inscriptions and labels of the axes in figures 9 and 11.
Commented [KM1]: You cited 12 studies, but you haven't provided thorough explanations for each reference. It would be helpful to add comments about each reference to provide more clarity.
Commented [KM6]: The latest version of HSC Chemistry is 10.3 - https://www.metso.com/globalassets/portfolio/hsc-chemistry/06-whats-new-in-hsc10.pdf?r=3
Commented [KM10]: I recommend adding your customized response within the text of the manuscript to provide further context and clarification.
Commented [KM11]: I believe that several explanations are necessary in the manuscript to improve understanding and provide more comprehensive information.
Author Response
Thank you very much for your review, below are our answers to your questions/concerns
It would be beneficial to enhance the inscriptions and labels of the axes in figures 9 and 11.
Thank you for your attention, the drawings have been corrected
Commented [KM1]: You cited 12 studies, but you haven't provided thorough explanations for each reference. It would be helpful to add comments about each reference to provide more clarity.
Thank you for your comment, we have provided thorough explanations for each reference
Commented [KM6]: The latest version of HSC Chemistry is 10.3 - https://www.metso.com/globalassets/portfolio/hsc-chemistry/06-whats-new-in-hsc10.pdf?r=3
Thank you for your attention. The error that occurred when entering the version has been corrected
Commented [KM10]: I recommend adding your customized response within the text of the manuscript to provide further context and clarification.
According to the reviewer's comment. this note has been added to the article.
Commented [KM11]: I believe that several explanations are necessary in the manuscript to improve understanding and provide more comprehensive information.
Thank you for your comment. In the attached document we send a detailed theoretical description of the phenomena. However, we recognize that this is too theoretical consideration to be suitable for inclusion in the article.
